# The Common and Unique Pattern of Microbiome Profiles among Saliva, Tissue, and Stool Samples in Patients with Crohn’s Disease

**DOI:** 10.3390/microorganisms10071467

**Published:** 2022-07-20

**Authors:** Seung Yong Shin, Sounkou Kim, Ji Won Choi, Sang-Bum Kang, Tae Oh Kim, Geom Seog Seo, Jae Myung Cha, Jaeyoung Chun, Yunho Jung, Jong Pil Im, Ki Bae Bang, Chang Hwan Choi, Soo-Kyung Park, Dong Il Park

**Affiliations:** 1Department of Internal Medicine, Chung-Ang University College of Medicine, Seoul 06973, Korea; mdthepage@gmail.com (S.Y.S.); gicch@cau.ac.kr (C.H.C.); 2Functional Genome Institute, PDXen Biosystems, Co., Suwon 16488, Korea; pdxenpurenine@gmail.com (S.K.); pdxen.jwchoi@gmail.com (J.W.C.); 3Department of Biological Sciences, Ulsan National Institute of Science and Technology, Ulsan 44919, Korea; 4Department of Biological Sciences, Sungkyunkwan University, Suwon 16419, Korea; 5Department of Internal Medicine, Daejeon St. Mary’s Hospital, The Catholic University of Korea College of Medicine, Daejeon 34943, Korea; dxandtx@catholic.ac.kr; 6Department of Internal Medicine, Haeundae Paik Hospital, Inje University College of Medicine, Busan 48108, Korea; kto0440@paik.ac.kr; 7Department of Gastroenterology and Digestive Disease Research Institute, Wonkwang University School of Medicine, Iksan 54538, Korea; medsgs@wonkwang.ac.kr; 8Department of Internal Medicine, Kyung Hee University Hospital at Gang Dong, Kyung Hee University College of Medicine, Seoul 05278, Korea; clicknox@khnmc.or.kr; 9Department of Internal Medicine, Gangnam Severance Hospital, Yonsei University College of Medicine, Seoul 06273, Korea; j40479@gmail.com; 10Division of Gastroenterology, Department of Medicine, Soonchunhyang University College of Medicine, Cheonan 31151, Korea; yoonho7575@naver.com; 11Department of Internal Medicine and Liver Research Institute, College of Medicine, Seoul National University, Seoul 03080, Korea; jp-im@hanmail.net; 12Department of Internal Medicine, Dankook University College of Medicine, Cheonan 31116, Korea; kibaebang@gmail.com; 13Division of Gastroenterology, Department of Internal Medicine and Inflammatory Bowel Disease Cencer, Kangbuk Samsung Hospital, Sungkyunkwan University School of Medicine, Seoul 03181, Korea; skparkmd@gmail.com; 14Medical Research Institute, Kangbuk Samsung Hospital, Sungkyunkwan University School of Medicine, Seoul 03181, Korea

**Keywords:** Crohn’s disease, microbiota, saliva, tissues, feces

## Abstract

This study aimed to elucidate common and unique microbiome patterns in saliva, intestinal tissue biopsy, and stool samples from patients with Crohn’s disease (CD). Saliva, tissue, and stool samples from patients with CD were prospectively collected. Quantitative and phylogenetic analyses of 16s rRNA sequencing data were performed with bioinformatical pipelines. A total of 30 patients were enrolled in this study. The composition of major microbial taxa was similar between tissue and stool samples. A total of 11 of the 20 most abundant microbiota were found in both samples. The microbial community in saliva was significantly distinct from that in tissue and stool. The major species of microbiota and their composition also differed significantly from those of tissue and stool samples. However, *Streptococcus* and *Prevotella* are common genera in saliva, tissue, and stool microbiome. The abundance of *Streptococcus*, *Pantoea*, and *Actinomyces* from the saliva sample group were significantly different, varying with the location of the inflammation. Saliva has a distinct microbial community compared with tissues and stools in patients with CD. *Prevotella* and *Streptococcus*, which are commonly observed in saliva, stool, and tissue, can be considered a potential biomarker related to the diagnosis or prognosis of CD.

## 1. Introduction

Inflammatory bowel disease (IBD) is a chronic inflammatory gastrointestinal disorder mainly represented by Crohn’s disease (CD) and ulcerative colitis (UC) [1,2]. The incidence and prevalence of IBD have increased worldwide, and patients with IBD exhibit various symptoms, including abdominal pain, diarrhea, and hematochezia. Despite continuous efforts, the etiology and pathogenesis of IBD remain poorly understood [3]. In recent years, the microbiome has become a major area of interest in the field of IBD research with the development of genome sequencing methods. Several studies have shown differences in the gut microbial community between patients with IBD and normal individuals, and increasing evidence suggests that they play a pivotal role in the pathogenesis of IBD [4,5]. Inflammatory cascades induced by an imbalance of the gut microbial community, called dysbiosis, are the main elements of this hypothesis [6].

Research on IBD through metagenome analysis has mainly focused on dysbiosis in the intestinal tissue or stool samples, areas that are in the vicinity of where the disease occurs, and discovered several microbiome biomarkers for IBD [1,2,3]. However, CD can affect any location in the gastrointestinal tract, from the oral cavity to the anus; therefore, the symptoms of CD are not specifically found within the small intestine and colon. It can also lead to disease in the joints, skin, liver, biliary ducts, and kidneys beyond the entire gastrointestinal tract, known as extraintestinal manifestations (EIMs) [7]. These bodily changes caused by CD make it possible to detect dysbiosis in sites other than intestinal tissue or stool samples [8].

Recently, several studies have investigated the oral microbiome in saliva samples from patients with CD based on disease characteristics [9,10]. Several oral mucosal diseases, including aphthous ulcers and stomatitis, are frequently found in patients with CD and are associated with disease activities [11,12]. Salivary sample collection has the advantage of being safer, easier, and less uncomfortable for patients compared to other samples. Thus, understanding the salivary microbiome will improve patient compliance and expand our view of microbiota in patients with CD. However, few studies on salivary microbiota have investigated the association between the microbiota in stool, intestinal tissue, and saliva in patients with CD. Therefore, this study aimed to elucidate the common and unique microbiome patterns in saliva, intestinal tissue biopsy, and stool samples from patients with CD.

## 2. Materials & Methods

### 2.1. Study Population

This prospective study was conducted at Kangbuk Samsung Hospital in Korea between May 2017 and December 2020. Patients aged >18 years who were diagnosed with CD and were undergoing treatment were included in this study. Baseline demographic and clinical characteristics were assessed before sample collection. Concomitant drug use was evaluated in each patient. Patients who had used probiotics and antibiotics that can affect the intestinal microbiome within the past three weeks were excluded. Participants refrained from smoking, not only for the case-control study but also as a part of the treatment. Patients who had used antacid that can affect the oral microbiome within the past three weeks were excluded. History of disease-related operations included incision and drainage of anal abscess, anal fistulectomy, small bowel resection, and ileocolectomy.

### 2.2. Sample Collection

Saliva, intestinal tissue, and stool samples were all taken from the same patients. Saliva samples (2 mL) were collected using a saliva collection kit (Cat. PDX-026; PDXen Biosystems Co., Daejeon, Korea), which can be transported and stored at room temperature (15–30 °C). The stool samples were collected by the participants (5 g each) and stored in a deep freezer (−80 °C) immediately after submission. Tissue biopsy samples were refrigerated at 4 °C immediately after being taken from intestine during endoscopy and stored at −80 °C. Five tissue samples collected at the ileocecal valve were analyzed in this study.

### 2.3. DNA Extraction and 16S rRNA Gene Sequence Processing

To separate the cellular pellet from the cell-free supernatant, the samples were subjected to centrifugation at 15,000 rpm for 20 min at 4 °C. DNA was extracted from the pellet using a QIAamp DNA Microbiome Kit (Qiagen, Valencia, CA, USA) following the manufacturer’s instructions.

The V3–V4 region of the 16S rRNA gene was amplified using 341F (5′-TCG TCG GCA GCG TCA GAT GTG TAT AAG AGA CAG CCT ACG GGN GGC WGC AG-3′) and 805R (5′-GTC TCG TGG GCT CGG AGA TGT GTA TAA GAG ACA GGA CTA CHV GGG TAT CTA ATC C-3′) primers with Illumina adaptor overhang sequences. Amplicons were purified with a magnetic bead-based clean-up system (Agencourt AMPure XP; Beckman Coulter, Brea, CA, USA). Indexed libraries were prepared by limited-cycle PCR using Nextera technology, further cleaned up, and pooled at equimolar concentrations. The final library was denatured with 0.2 N NaOH and diluted to 6 pM with a 20% PhiX control. Sequencing was performed on the Illumina MiSeq platform using a 2 × 300 bp paired-end protocol, according to the manufacturer’s instructions

### 2.4. Bioinformatics and Statistical Analysis

The primary analysis of the obtained sequences was performed using demultiplexing with MiSeqReporter software (Illumina). The paired-end sequences of each sample were exported from the MiSeq system for analysis in the FASTA format.

Quantitative analysis of sequence reads was conducted using Quantitative Insights Into Microbial Ecology 2 (QIIME2) 2021.4.0 [13]. The demultiplexed sequences were denoised using DADA2 [14] (via q2-dada2). We aligned amplicon sequence variants (ASVs) through an align-to-tree-mafft-fasttree pipeline consisting of q2-alignment MAFFT [15], and q2-phylogeny FasTtree2 [16] provided by QIIME2.

Alpha diversity metrics (observed features, Faith’s phylogenetic diversity [17], evenness, chao1, and Shannon entropy) and beta diversity metrics (weighted UniFrac [18], unweighted UniFrac [19], Jaccard distance, and Bray–Curtis dissimilarity) were estimated using q2-diversity after samples were rarefied (subsampled without replacement), with 10,954 sequences per sample.

Taxonomy was assigned to ASVs using the q2-feature-classifier [20] classify-sklearn naïve Bayes taxonomy classifier against the Silva 138 99% operational taxonomic units (OTUs) full-length sequences [21]. We visualized differences in the microbiome profiles of saliva, stool, and tissue samples using Uniform Manifold Approximation and Projection (UMAP) [22] and Linear Discriminant Analysis (LDA) effect size (LEfSe) [23].

## 3. Results

### 3.1. Baseline Characteristics

A total of 30 patients were enrolled in the study. Table 1 shows the baseline demographic and clinical characteristics of all participants. The mean patient age was 35.7 years, and 70.0% of enrolled patients were male. The mean disease duration was 6.8 years, and the mean CDAI was 52.7. A total of 53.4% of patients received biologic agents.

Demultiplexed sequence counts were distributed as follows: minimum, 30,800; median, 101,197; mean, 159,663; and maximum, 1,270,567; totaling 14,369,673. DADA2 trimming was performed using the standard Phred quality score 35 for the 5′ end and 20 for the 3′ end for each direction of sequence reads; 14–299 positions in forward reads and 5–226 positions in reverse reads. Nine samples (seven from tissue and two from stool) were filtered out before alpha and beta diversity analyses because they had fewer feature counts than sampling depth, 10,954, determined based on the balance of sufficient level of rarefaction and minimizing sample loss.

### 3.2. Diversity in Microbiota

Nine samples (seven from tissue, two from stool) were filtered out only when alpha and beta diversity analysis were performed due to fewer feature counts than sampling depth, 10,954, determined based on the balance of an adequate level of rarefaction and minimizing sample loss.

#### 3.2.1. Alpha Diversity

The tissue sample group showed the most varied result in Shannon entropy and Chao1 indexes, with widely spread values (the highest maximum (over 3000) and median in Chao1 and the highest maximum and median in Shannon entropy). The Kruskal–Wallis *p*-value was 1.85e-8 for Shannon entropy and 1.57 × 10^−9^ for Chao1, which indicates a statistically significant difference in the median of each sample group (Figure 1A).

#### 3.2.2. Beta Diversity

Boxplots of unweighted UniFrac (Figure 1B–D), incorporating distances in phylogenic tree among members in comparing groups, also showed similar results for alpha diversity. Higher internal distances (mean: saliva, 0.45; stool, 0.5; tissue, 0.87) of members and the highest distance between groups (mean: saliva–stool, 0.66; saliva–tissue, 0.9; stool–tissue, 0.93) were found in the tissue sample group. Overall distribution patterns of weighted UniFrac are similar to the unweighted version, while distances within the tissue sample group are varied (Appendix A).

### 3.3. Taxonomy Distribution

These results of the diversity analysis agreed with the taxonomic distribution plotted in Figure 2; the most abundant OTU in the tissue sample group was ‘unassigned’, indicating a mixture of unidentified bacteria. Seven major OTUs and the ‘unassigned’ group are listed.

### 3.4. Cluster Visualization

Normalized abundances of each OTU among samples were projected on UMAP (Figure 3A) and a heatmap (Figure 3B) based on values that were calculated by the natural log of raw OTU counts plus one pseudo count, resulting in a clearly separated saliva sample group from stool and tissue groups. The composition of the strains of OTU were appropriate to their environmental conditions; tissue and stool sample groups neighboring each other in the intestines were similar, but different from the saliva sample group from the oral cavity.

As shown in Figure 3A, the tissue sample group was divided into major subgroups near the stool sample group and minor subgroups isolated from any other groups. The separated location of two subgroups of the tissue sample group can explain the result of UniFrac distance in Figure 1C,D. The minor subgroup makes the distance from saliva closer than from stool, which seems to contradict the fact that the distance from saliva is more distant in tissue than stool. It also explains the varied distances within the tissue sample group in weighted UniFrac distance (Appendix A). Nine members of the minor subgroup had a high abundance of *Streptococcus* in the microbiome, and most (seven out of nine, 77.8%) were female.

### 3.5. Quantitative and Phylogenic Analysis

The histogram of LEfSe results shows OTUs with an LDA score over 4.0, and *p*-values of the Kruskal–Wallis test and Wilcoxon test were less than 0.05 for each group (Figure 4A). ‘Unassigned’, which takes the most remarkable portion of the tissue samples in this histogram, shows it affected various other results. *Streptococcus*, *Serratia*, and *Prevotella* in saliva and *Escherichia-Shigella* and *Bacteroides* genera in stool were marked. These genera were also marked as representative identities for the upper families—*Streptococcaceae*, *Prevotellaceae*, and *Bacteroidaceae*—in the cladogram (Figure 4B).

The 20 most abundant genera in the saliva, stool, and tissue are listed in Table 2A–C in order of relative abundance. Figure 5 shows a Venn diagram of these 20 genera with an inclusion relationship, and the names of the elements are listed in Table 3. *Prevotella* and *Streptococcus* were common among the three sample types. *Fusobacterium*, *Actinomyces*, *Rothia*, and *Veiollonella* were found in common in saliva and tissue samples.

### 3.6. Clinical Subgroup Analysis

Several clinical aspects based on the Montreal Classification [24], such as sex, age group, behavior, and location of inflammation, were tested using the Kruskal–Wallis test among sample groups (Table 4). Several genera showed a different abundance in each subgroup. In particular, the abundance of *Streptococcus*, *Pantoea*, and *Actinomyces* in the saliva sample group were significantly different (*p*-value < 0.05) among the locations of inflammation. The stool sample group showed *Megamonas* and *Collinsella* as significant genera under these conditions, and there were no significant genera in the tissue sample group. In contrast, five and four genera in the stool and tissue sample groups, respectively, were significantly found in the behavioral aspect, where no genus was found in the saliva sample group.

## 4. Discussion

This study observed taxonomic distributions in the saliva, tissue, and stool samples of patients with CD. We employed saliva samples from the oral cavity for our analysis for the following reasons: (1) several types of research describe the composition of the microbiome of saliva affected by inflammation, and (2) it is much easier to collect samples from patients in terms of time, occasion, and quality control compared to intestinal tissue biopsy or stool. Due to these advantages, we examined the plausibility of suggesting the saliva microbiome from the oral cavity as a diagnostic material for CD through comparison and contrast with the microbiomes of stool and intestinal tissue biopsy.

The bacterial community of the intestinal tissue and stool samples was similar. They were not significantly separated in UMAP. Of the top 20 microbiota, 11 were common in both the tissue and stool samples. However, as expected, the microbial community in saliva was significantly distinct from that in tissue and stool. As shown in Table 2, the major species of microbiota and their compositions were also significantly different. However, several species accounted for a relatively large proportion in each sample. *Prevotella* and *Streptococcus* were commonly observed in saliva, stool, and tissue.

*Streptococcus* and *Prevotella* are considered to be the main species associated with dysbiosis in the salivary microbiota of patients with IBD [25,26]. *Prevotella*, a Gram-negative obligate anaerobe, is a commensal microbiota prevalent in the gastrointestinal tract, from the oral cavity to the anus. Previous studies reported a decreased abundance of *Streptococcus* and an increased abundance of *Prevotella* in patients with IBD compared to healthy controls, suggesting that these changes in oral microbiota may be associated with immune disorders in the pathogenesis of IBD [9]. However, the role of *Prevotella* in human health and disease remains unclear. An increased abundance of *Prevotella* species in the gut has been reported in various inflammatory diseases, such as bacterial vaginosis, rheumatoid arthritis, esophagitis, and gastritis [27,28,29]. Although not yet fully established, prior studies have noted that *Prevotella* species are involved in several inflammatory cascades [27]. *Prevotella* activates Toll-like receptor 2, which drives the immune response by producing inflammatory cytokines such as interleukin-23 (IL-23). Another pathway is the direct stimulation of epithelial cells to produce cytokines that promote the T helper type 17 (Th17)-associated immune response and neutrophil recruitment. Adhering to the host cell membrane and destroying it through bacterial substances has also been suggested as an inflammatory reaction [30]. It has been shown that some species of *Prevotella* induce colitis in mice [27], and an increased abundance of *Prevotella* in colonic biopsy samples was identified in patients with IBD compared to that in healthy controls [31]. Salivary *Prevotella* is an oral biofilm-forming bacterium and has commonly been reported as one of the main taxa that are increased in abundance in patients with IBD [9,32,33]. They are also correlated with inflammatory biomarkers, including immunological cytokines [9,32]. Recently, it was suggested that salivary *Prevotella* might serve as a biomarker in patients with CD [9]. Our study is consistent with previous research and demonstrates that *Prevotella* is a universally abundant taxon in the saliva, intestinal tissue, and stool of patients with CD. Interestingly, the relative abundance of *Prevotella* in salivary samples was prominent compared to that in tissue and stool samples. This result indicates a potential role of salivary *Prevotella* in patients with CD.

*Streptococcus* is an oral commensal bacterium and is the most-abundant bacteria forming dental biofilm with the ability to bind and coaggregate with other microorganisms using surface molecules [34]. However, some species of *Streptococcus* are known to have anti-inflammatory effects and are considered oral protective probiotics [35]. The depletion of salivary *Streptococcus* in patients with IBD was found in previous research, and the abundance of salivary *Streptococcus* was reported to be negatively correlated with serum inflammatory markers, such as white blood cells and C-reactive protein [9], as well as inflammatory cytokines [9,32]. Similar to our study, one of the *Streptococcus* species were found in both stool and saliva samples in previous studies [36]. The abundance of *Streptococcus* in saliva samples also significantly differed among the locations of inflammation in our study. A further study with more focus on *Streptococcus* in the saliva of patients with CD can be considered.

*Fusobacterium* is commonly found in saliva and stool samples. *Fusobacterium* was reported as a salivary indicator in patients with CD [37,38]. Most bacterial species, including *Prevotella*, *Streptococcus*, and *Fusobacterium*, identified in common between samples are already known to have compositional changes in saliva, tissue, and stool samples in patients with IBD. It is noteworthy that these bacteria overlapped between samples. Further studies with a great focus on this result are recommended.

This study has several limitations. First, the relatively small sample size may have been a potential for bias. Second, most patients had mild disease status when the samples were collected. Third, factors affecting the microbiome, such as diet, were not controlled for before sample collection. Fourth, the results of this study do not explain the correlation between salivary and stool or tissue microbiota. Fifth, the tissue microbiome may be different depending on the presence or absence of inflammation. We judged that the difference of location would be a higher risk of bias than the presence or absence of inflammation, based on previous studies [39,40,41], and analyzed only samples collected from the ileocecal valve. In addition, they would not have had much effect on the outcome because most patients (95% or more) had no inflammation of the ICV. Finally, this research only included patients with CD without comparable healthy controls. Therefore, these results should be interpreted with caution, and for further understanding and clinical application, studies are needed to compare with healthy controls. However, this study was intended to investigate the relationship between saliva, tissue, and stool microbiome within patients with CD rather than comparing healthy control with patients with CD, and only a few studies have evaluated differences in the microbiomes of saliva, stool, and tissue in patients with IBD. We analyzed all saliva, stool, and tissue samples to demonstrate their differences and identify bacteria commonly found between samples. Further large-scale studies that consider the factors associated with the microbial community are required to validate our findings.

## 5. Conclusions

In this study, we approached CD through microbiota from the oral cavity and intestines, by analyzing saliva, tissue biopsy, and stool samples. We found *Streptococcus* and *Prevotella* to be the most abundant and statistically significant overlapping genera among the microbiome from saliva, tissue biopsy, and stool samples. At the same time, saliva had unique microbial characteristics compared to the other two samples, implying it to be a new basis for exploring candidates for CD biomarkers. Since it is easy to obtain and manage saliva samples from patients, further research examining the common genera that might be diagnostic or prognostic biomarkers would improve patient care and expand the knowledge of clinicians and doctors dealing with patients with CD.

There are still many unanswered questions in the field of microbiomes of patients with IBD. This study could help expand our understanding of the microbiota in patients with CD and is expected to be used as data for future studies to develop a complete picture of the microbiome in CD.

## Figures and Tables

**Figure 1 microorganisms-10-01467-f001:**
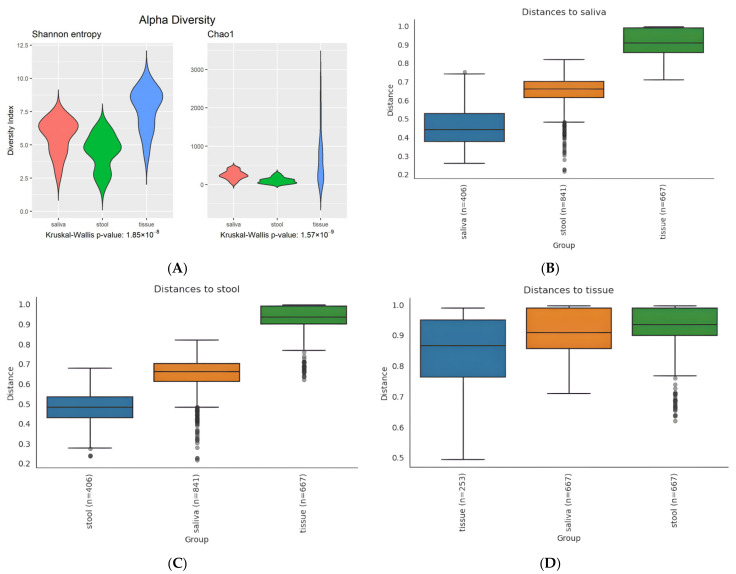
Diversity in microbiota. Alpha diversity of the microbiome in saliva, tissue, and stool samples of patients with CD (**A**). Beta diversity; unweighted UniFrac distance to saliva (**B**), stool (**C**), and tissue (**D**) of microbiome in the saliva, tissue, and stool of CD patients. Unweighted UniFrac distance incorporates distances in the phylogenic tree among members in comparing groups. n, match count for each cases; the saliva (29)–stool (29) pair has 841 (=29 × 29) matches, and tissue (23)–saliva or –stool (29) have 667 (=23 × 29). Cases within groups are 406 (=29C2) for saliva or stool and 253 (=23C2) for tissue. PERMANOVA resulted in a *p*-value of 0.001 (<0.05) for all three pair cases, indicating each pairwise cases have statistically significant difference.

**Figure 2 microorganisms-10-01467-f002:**
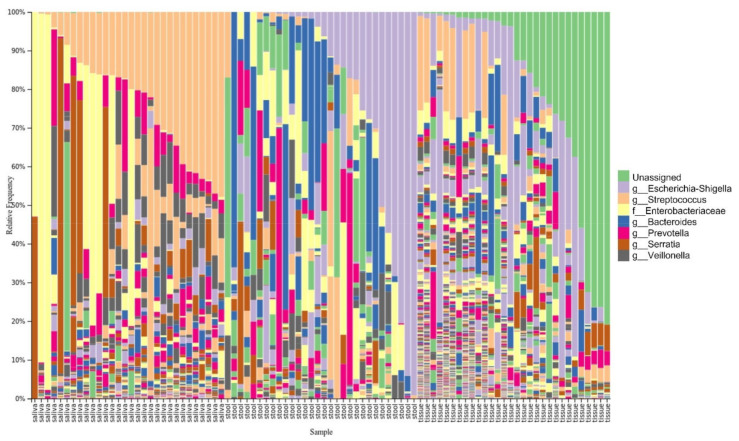
Taxonomy bar plot of the microbiome in the saliva, tissue, and stool samples of patients with CD.

**Figure 3 microorganisms-10-01467-f003:**
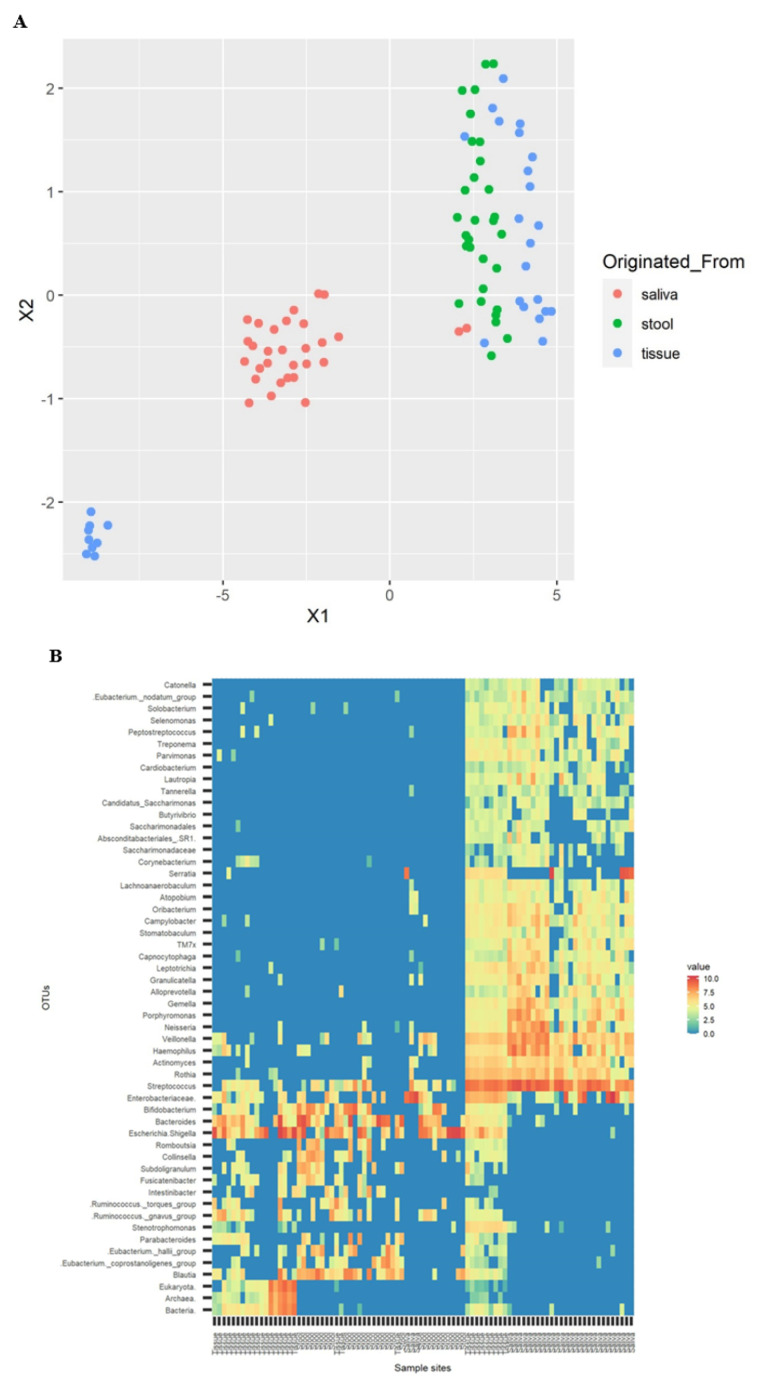
UMAP of the microbiome cluster in the saliva, tissue, and stool samples of patients with CD (**A**). Heatmap of the microbiome cluster in the saliva, tissue, and stool samples of patients with CD (**B**).

**Figure 4 microorganisms-10-01467-f004:**
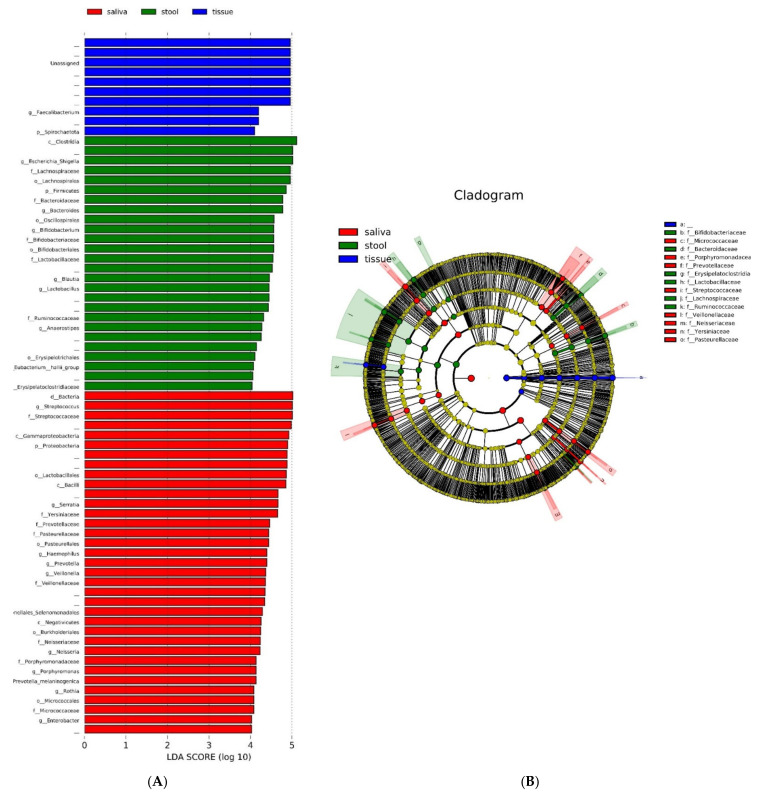
LDA histogram of the microbiome in the saliva, tissue, and stool samples of patients with CD (**A**) and a cladogram of the microbiome in the saliva, tissue, and stool samples of patients with CD (**B**). *Streptococcus*, *Serratia*, and *Prevotella* in saliva and *Escherichia-Shigella* and *Bacteroides* genera in stool were marked. These genera were also marked as representative identities for the upper families—*Streptococcaceae*, *Prevotellaceae*, and *Bacteroidaceae*—in the cladogram.

**Figure 5 microorganisms-10-01467-f005:**
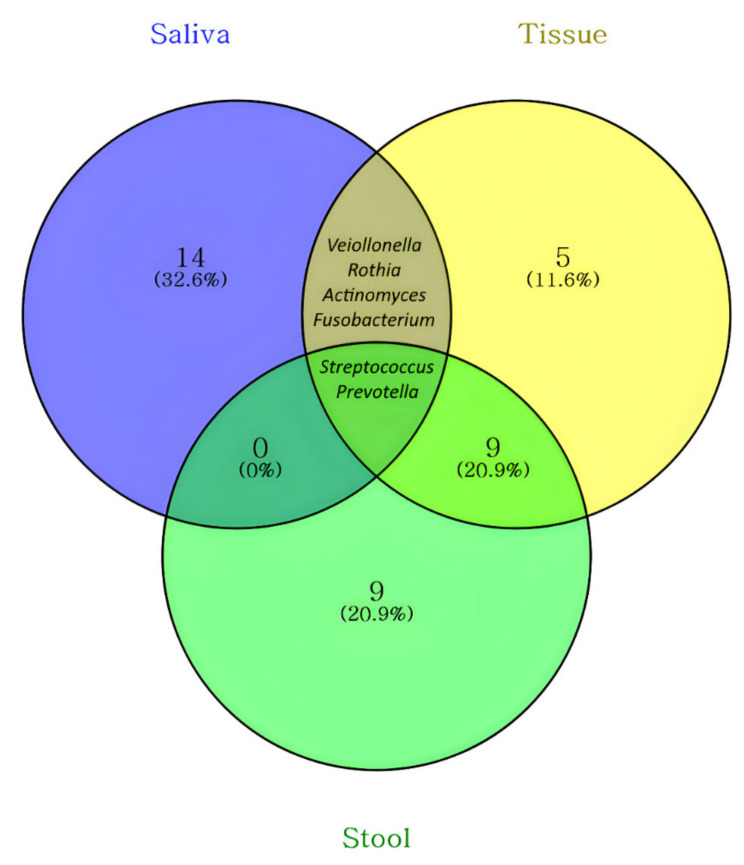
Venn diagram of top twenty abundant genera in the microbiome in the saliva, tissue, and stool samples of patients with CD. *Prevotella* and *Streptococcus* were common among the three sample types.

**Table 1 microorganisms-10-01467-t001:** Baseline demographic and clinical characteristics of participants.

	CD (n = 30)
Age (year) mean ± SD	35.7 ± 11.2
Male, n (%)	21 (70.0)
BMI (kg/m^2^), mean ± SD	22.4 ± 4.4
Smoking status, n (%)	
Current	4 (13.3)
Former	1 (3.3)
Never	25 (83.3)
Unknown	0 (0.0)
Disease duration (year), mean ± SD	6.8 ± 6.0
Disease location, n (%)	
Ileum (L1)	4 (13.3)
Colon (L2)	8 (26.7)
Ileocolonic (L3)	16 (53.3)
Ileocolonic (L3) + upper GI (L4)	2 (6.7)
CDAI, mean ± SD	52.7 ± 57.5
Extraintestinal manifestation, n (%)	
Arthritis/arthralgia	4 (13.3)
Uveitis/iritis	0 (0.0)
Reactive skin lesion	0 (0.0)
Concomitant drug use, n (%)	
5-ASAs	22 (73.3)
Corticosteroid	16 (53.3)
Azathioprine/6-mercaptopurine	0 (0.0)
Infliximab	14 (46.7)
Adalimumab	0 (0.0)
Ustekinumab	2 (6.7)
Previous history of disease-related operations	11 (36.7)

SD, standard deviation; BMI, body mass index; CDAI, Crohn’s disease activity index; 5-ASA, 5-aminosalicylic acid.

**Table 2 microorganisms-10-01467-t002:** (**A**) Top 20 abundant genera in the saliva sample. (**B**) Top 20 abundant genera in the tissue sample. (**C**) Top 20 abundant genera in the stool sample.

(A)		
Genus	Rate of Containing Samples ^a^	Total Relative Abundance ^b^
*Streptococcus*	96.67%	22.29%
*Serratia*	26.67%	9.61%
*Prevotella*	100.00%	6.00%
*Veillonella*	93.33%	5.26%
*Haemophilus*	83.33%	4.83%
*Neisseria*	83.33%	3.43%
*Porphyromonas*	86.67%	2.63%
*Rothia*	90.00%	2.45%
*Pantoea*	3.33%	2.18%
*Enterobacter*	16.67%	2.07%
*Actinomyces*	93.33%	1.74%
*Gemella*	90.00%	1.68%
*Fusobacterium*	83.33%	1.58%
*Campylobacter*	86.67%	1.21%
*Leptotrichia*	86.67%	1.00%
*Peptostreptococcus*	76.67%	0.99%
*Granulicatella*	80.00%	0.96%
*Alloprevotella*	80.00%	0.91%
*TM7x*	76.67%	0.88%
*Capnocytophaga*	86.67%	0.77%
^a^ Proportion of samples where one or more OTU count is detected. ^b^ Means of relative abundances of each OTU for samples.		
**(B)**		
**Genus**	**Rate of Containing Samples**	**Total Relative Abundance**
*Escherichia Shigella*	93.33%	13.47%
*Streptococcus*	70.00%	6.71%
*Bacteroides*	90.00%	3.91%
*Faecalibacterium*	73.33%	2.95%
*Anaerostipes*	66.67%	2.49%
*Brachyspira*	3.33%	2.00%
*Ruminococcus gnavus group*	76.67%	1.46%
*Prevotella*	66.67%	1.41%
*Ruminococcus torques group*	46.67%	1.14%
*Lachnoclostridium*	63.33%	1.08%
*Rothia*	40.00%	1.08%
*Veillonella*	46.67%	0.97%
*Megamonas*	36.67%	0.94%
*Sutterella*	40.00%	0.87%
*Blautia*	70.00%	0.77%
*Bifidobacterium*	76.67%	0.77%
*Clostridium sensu stricto 1*	56.67%	0.75%
*Fusobacterium*	63.33%	0.72%
*Actinomyces*	33.33%	0.71%
*Pseudomonas*	53.33%	0.69%
**(C)**		
**Genus**	**Rate of Containing Samples**	**Total Relative Abundance**
*Escherichia Shigella*	80.00%	20.80%
*Bacteroides*	73.33%	12.04%
*Bifidobacterium*	63.33%	7.04%
*Lactobacillus*	33.33%	5.69%
*Blautia*	70.00%	5.63%
*Anaerostipes*	46.67%	4.17%
*Faecalibacterium*	46.67%	2.29%
*Eubacterium hallii group*	40.00%	2.23%
*Collinsella*	40.00%	1.91%
*Lachnoclostridium*	43.33%	1.88%
*Streptococcus*	56.67%	1.79%
*Eubacterium coprostanoligenes group*	40.00%	1.78%
*Prevotella*	26.67%	1.73%
*Megasphaera*	16.67%	1.72%
*Megamonas*	10.00%	1.44%
*Ruminococcus gnavus group*	36.67%	1.31%
*Romboutsia*	36.67%	1.30%
*Morganella*	6.67%	1.20%
*Pediococcus*	10.00%	1.09%
*Subdoligranulum*	30.00%	1.08%

**Table 3 microorganisms-10-01467-t003:** Common and exclusive genera among saliva, tissue, and stool samples.

Common Sites	Genus	Exclusive Sites	Genus
Saliva, tissue, and stool	*Prevotella*	Saliva	*Alloprevotella*
	*Streptococcus*		*Campylobacter*
Saliva and tissue	*Actinomyces*		*Capnocytophaga*
	*Fusobacterium*		*Enterobacter*
	*Rothia*		*Gemella*
	*Veillonella*		*Granulicatella*
Tissue and stool	*Ruminococcus gnavus group*		*Haemophilus*
	*Anaerostipes*		*Leptotrichia*
	*Bacteroides*		*Neisseria*
	*Bifidobacterium*		*Pantoea*
	*Blautia*		*Peptostreptococcus*
	*Escherichia Shigella*		*Porphyromonas*
	*Faecalibacterium*		*Serratia*
	*Lachnoclostridium*		*TM7x*
	*Megamonas*	Tissue	*Ruminococcus torques group*
			*Brachyspira*
			*Clostridium sensu stricto 1*
			*Pseudomonas*
			*Sutterella*
		Stool	*Eubacterium coprostanoligenes group*
			*Eubacterium hallii group*
			*Collinsella*
			*Lactobacillus*
			*Megasphaera*
			*Morganella*
			*Pediococcus*
			*Romboutsia*
			*Subdoligranulum*

**Table 4 microorganisms-10-01467-t004:** *p*-values of Kruskal–Wallis test by sample groups among clinical subgroups (continued to next page).

Site	Genus	Sex	Behavior (1,2,3) ^a^	Behavior (Perianal) ^b^	Age Group ^c^	Location (2 vs. 1, 3) ^d^	Location (1,2,3) ^e^
Saliva	*Streptococcus*	2.30 × 10^−1^	7.39 × 10^−1^	8.35 × 10^−1^	2.71 × 10^−1^	2.16 × 10^−2 f^	5.84 × 10^−2^
	*Veillonella*	9.10 × 10^−1^	5.44 × 10^−1^	3.29 × 10^−1^	1.99 × 10^−2 f^	9.25 × 10^−1^	3.71 × 10^−1^
	*Pantoea*	1.27 × 10^−1^	5.65 × 10^−1^	2.85 × 10^−1^	4.14 × 10^−1^	5.46 × 10^−1^	3.88 × 10^−2 f^
	*TM7x*	4.80 × 10^−1^	5.80 × 10^−1^	6.45 × 10^−1^	3.69 × 10^−2 f^	1.00 × 10^−0^	8.80 × 10^−1^
	*Actinomyces*	6.19 × 10^−1^	7.81 × 10^−1^	8.52 × 10^−1^	3.10 × 10^−1^	1.79 × 10^−2 f^	4.80 × 10^−2 f^
Stool	*Escherichia.Shigella*	3.18 × 10^−1^	1.48 × 10^−2 f^	1.94 × 10^−2 f^	1.81 × 10^−2 f^	5.40 × 10^−1^	7.09 × 10^−1^
	*Bacteroides*	3.61 × 10^−1^	4.18 × 10^−2^	7.53 × 10^−1^	2.77 × 10^−1^	9.25 × 10^−1^	1.92 × 10^−1^
	*Bifidobacterium*	7.63 × 10^−1^	2.89 × 10^−2^	6.70 × 10^−1^	8.22 × 10^−1^	1.36 × 10^−1^	2.27 × 10^−1^
	*Megamonas*	8.28 × 10^−1^	1.60 × 10^−1^	5.54 × 10^−2^	1.76 × 10^−1^	2.96 × 10^−3 f^	1.21 × 10^−2 f^
	*Eubacterium coprostanoligenes group*	8.69 × 10^−2^	9.53 × 10^−1^	1.52 × 10^−1^	2.38 × 10^−2 f^	8.12 × 10^−1^	5.24 × 10^−1^
	*Megasphaera*	7.01 × 10^−1^	3.21 × 10^−1^	2.50 × 10^−2 f^	6.84 × 10^−1^	3.48 × 10^−1^	5.32 × 10^−1^
	*Lachnoclostridium*	7.44 × 10^−3 f^	1.69 × 10^−1^	2.32 × 10^−1^	2.63 × 10^−1^	3.51 × 10^−1^	3.04 × 10^−1^
	*Eubacterium hallii group*	3.19 × 10^−1^	5.48 × 10^−1^	2.15 × 10^−2 f^	8.82 × 10^−1^	4.27 × 10^−1^	2.35 × 10^−1^
	*Collinsella*	7.59 × 10^−1^	5.04 × 10^−1^	3.73 × 10^−1^	4.41 × 10^−1^	2.62 × 10^−2 f^	7.95 × 10^−2^
Tissue	*Streptococcus*	1.04 × 10^−3 f^	4.75 × 10^−1^	1.29 × 10^−2^	2.12 × 10^−1^	5.37 × 10^−1^	1.76 × 10^−1^
	*Prevotella*	6.51 × 10^−3 f^	8.21 × 10^−1^	5.67 × 10^−1^	2.62 × 10^−1^	2.14 × 10^−1^	4.55 × 10^−1^
	*Veillonella*	1.09 × 10^−3 f^	8.71 × 10^−1^	2.41 × 10^−1^	6.85 × 10^−1^	1.40 × 10^−1^	1.96 × 10^−1^
	*Rothia*	4.64 × 10^−4 f^	6.49 × 10^−1^	2.60 × 10^−1^	1.88 × 10^−1^	6.34 × 10^−1^	6.08 × 10^−1^
	*Faecalibacterium*	8.02 × 10^−1^	2.54 × 10^−2 f^	1.93 × 10^−1^	5.20 × 10^−1^	6.83 × 10^−2^	1.45 × 10^−1^
	*Clostridium sensu stricto 1*	3.57 × 10^−2 f^	5.72 × 10^−1^	2.98 × 10^−1^	5.82 × 10^−1^	2.61 × 10^−1^	4.43 × 10^−1^
	*Megamonas*	1.35 × 10^−1^	2.29 × 10^−1^	4.85 × 10^−2 f^	5.30 × 10^−2^	5.50 × 10^−1^	2.61 × 10^−1^
	*Actinomyces*	1.43 × 10^−4 f^	9.52 × 10^−1^	2.76 × 10^−1^	2.60 × 10^−1^	1.89 × 10^−1^	1.06 × 10^−1^
	*Pseudomonas*	2.07 × 10^−3 f^	7.02 × 10^−1^	2.26 × 10^−2 f^	3.41 × 10^−1^	2.35 × 10^−1^	2.40 × 10^−1^

^a^ Compared subgroups of each sample group by the behavior of CD; B1 (nonstricturing nonpenetrating), B2 (stricturing), and B3 (penetrating). ^b^ Compared subgroups of each sample group by the behavior of CD; perianal and non-perianal. ^c^ Compared subgroups of each sample group by the age group of patients; A1 (≤16 years), A2 (17–40 years), and A3 (>40 years). There were only A2 and A3 groups in samples. ^d^ Compared subgroups of each sample group by the location of CD; L2 (colon) against L1 (terminal ileum) and L3 (ileocolon). ^e^ Compared subgroups of each sample group by the location of CD; L1, L2, and L3 each. ^f^
*p*-value < 0.05.

## Data Availability

The data underlying this article will be shared on reasonable request to the corresponding author.

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
