# Peer review of "The Common and Unique Pattern of Microbiome Profiles among Saliva, Tissue, and Stool Samples in Patients with Crohn’s Disease"

_microorganisms, 2022, doi:10.3390/microorganisms10071467_

Round 1

Reviewer 1 Report

-          “are the mainelements of this hypothesis”

Correct the typo

-          “These bodily changes caused by CD make it possible to detect dysbiosis in sites other than intestinal tissue or stool samples.”

Add a citation (“Baima G et al. Shared microbiological and immunological patterns in periodontitis and IBD: A scoping review. Oral Dis. 2022 May;28(4):1029-1041. doi: 10.1111/odi.13843.”)

-          How did you choose the sample size?

Reviewer 2 Report

The manuscript by Seung Yong Shin et al., compares microbiome patterns in saliva, stool and intestinal tissue sample in patient with Chron´s disease. There were preformed quantitative and phylogenetic of analyses of 16s rRNA sequencing in 30 patients. They found the microbiome pattern in saliva was significantly distinct from that found in tissue and stool. The sequencing analysis of microbiome is well used just  the design and analysis of this research  is poor.

Why the sample of tissue, tool, saliva of patient is not compared to healthy sample. This comparison between disease sample in different tissue is very difficult to understand. In the sample selection there is too much diversity in individual parameter. Too many diversities between age range 35+/-11 treatment with corticosteroid 15 of 30 patient (50% of population analyzed) 16 patients, disease duration 6.8+/-6.0. Due to this variety in the input sample, it is difficult to do some general conclusion that could explain changes in microbiota due to Chron`s disease. Could you group 30 patient in two groups for example on cortiosteriode drugs and the ones without drugs  or old and young patients and compare their microbiome profile. At least you could pull out some information from this kind of design of sample you are comparing.

The title of figure 1 is misleading it start with alpha diversity although the figure is also about beta diversity, better title of this figure 1 would be have a title; “Diversity in microbiota” Then labeling of the chapters under chapter 2 is confusing. It should be labeled 1.1 and 1.2.

Reviewer 3 Report

The manuscript was reviewed for publication in the journal. The manuscript was designed to evaluate common and unique microbiome pattern in saliva, intestinal tissue biopsy, and stool samples from patients with Crohn’s disease (CD). The results obtained show that Streptococcus and Prevotella may be the most abundant and statistically significant overlapping genera among the microbiome from saliva tissue biopsy, and stool samples, and that saliva has unique microbial characteristic compared to the other two samples. It is the reviewer’s opinion that the manuscript is interesting and easy to follow. However, it appears that there are a couple of concerns in the manuscript.

1) Table 1 shows baseline demographic and clinical characteristics of participants. In the table, disease location includes ileum, colon, and ileocolonic. Under the different disease locations, where was the location that the intestinal tissue sample was collected? If the intestinal tissue sample was collected in the same location for all patients, the microbiome could be affected by the disease location. The authors should explain/discuss the issue.

2) The microbiome in saliva is considered to be sensitive to gastric acid. How about the use of antacid in patients? Patients who had used antacid should be excluded? The authors should discuss the issue.

3) Figure 1B, C and D show beta diversity (unweighted UniFrac distance). Why did the authors not show weighted UniFrac distance? 

4) The authors have discussed the limitation in the study, but healthy controls may be required to understand the results.

Round 2

Reviewer 3 Report

The manuscript was re-reviewed for publication in the journal. The manuscript was designed to evaluate common and unique microbiome pattern in saliva, intestinal tissue biopsy, and stool samples from patients with Crohn’s disease (CD). It is the reviewer’s opinion that the manuscript is interesting and easy to follow. The authors promptly explained/discussed the issues suggested. However, it appears that there is still a concern in the manuscript.

1) The authors mentioned that five samples were collected from ileocecal valve in all patients. In some patients, the ileocecal valve is considered to have an inflammation of CD. Also, the microbiomes between non-inflammation site of ileocecal valve and the inflammation site appear to be different. Therefore it appears to be questionable that five samples were collected from ileocecal valve in all patients. The authors should discuss/explain the issue.
